# Maneuverability Performance of a KRISO Container Ship (KCS) with a Bulb-Type Wavy Twisted Rudder and Asymmetric Pre-Swirl Stator

Yong-Jin Shin [1], Moon Chan Kim [2,*], Kyuong-Wan Lee [3], Woo Seok Jin [2] and Jin Wook Kim [2]

1    Samsung Heavy Industries Co., Ltd., Daejeon 34051, Republic of Korea; prop.shin@samsung.com
2    Department of Naval Architecture and Ocean Engineering, Pusan National University,
     Busan 46421, Republic of Korea; jws0537@pusan.ac.kr (W.S.J.); jinwookkim@pusan.ac.kr (J.W.K.)
3    Technology Part Blue Marine, Busan 46726, Republic of Korea; skylkw@naver.com
*    Correspondence: kmcprop@pusan.ac.kr; Tel.: +82-51-510-2401

**Abstract:** The maneuverability performance of a KCS with energy-saving devices was investigated. A bulb-type wavy rudder and an asymmetric pre-swirl stator were used as energy-saving devices because their use resulted in considerable efficiency gain. A validated simulation method based on the maneuvering modeling group model was used for the simulation in this study. Turning circle, 10/10 zigzag, and 20/20 zigzag tests were simulated to compare various cases with and without a stator as well as conventional full-spade and bulb-type wavy rudders. Remarkable maneuverability performance was achieved, particularly with a bulb-type wavy rudder, and starboard–port unbalancing due to the one-directional propeller action was almost eliminated using these devices. The findings of this study will be useful in the development of more reliable autonomous maritime surface ships.

**Keywords:** combined effect of ESD and rudders; wavy twisted rudder; asymmetric pre-swirl stator; maneuverability; efficiency





## 1. Introduction

In recent years, human damage and marine pollution due to collisions and ship grounding have become a serious problem. For instance, the Suez Canal grounding accident in 2021 caused significant economic and environmental losses. Maritime autonomous surface ships [1,2] have been a hot topic in the field of naval architecture in recent years. Maneuverability is essential for the development of such ships, particularly because reliability is the most important factor. Special rudders, such as flaps and schilling rudders [3–5], are being developed and applied to improve ship maneuverability. Special rudders are known to provide excellent maneuverability by delaying stall and generating high lift, but in actual operating conditions, efficiency loss due to additional drag and maintenance difficulties due to complex structures are serious problems.

In terms of rudder development, the semi-spade rudder has been improved to a full-spade rudder by removing the gap that causes cavity damage. Full-spade rudders are mainly used in container ships these days, which are prone to erosion by cavitation. The twist rudder has been developed by twisting the upper and lower parts of the full-spade rudder to accommodate the inflow from the propeller, in order to improve efficiency as well as cavitation performance. Shen, Jiang, and Remmers [6] confirmed this improvement through an experiment. Although the twisted rudder is more favorable to incoming flow, there is still some cavity risk in the twisted part. To solve this problem, a rudder bulb is adopted in the twisted part. Yoon et al. [7] conducted studies on the waveform in a uniform flow and confirmed that a high lift force is generated by delaying the stall. The twisted rudder concept has been applied to improve the performance of this wavy rudder. Because

of the overall smooth surface with a wavy configuration, a wavy twisted rudder is a better option for discontinuity in the twisted part compared with a rudder bulb. In addition, the wavy twisted rudder has better maneuverability than the conventional twisted rudder, whereas its efficiency is similar or slightly lower. The wavy rudder has been improved by aligning the wavy configuration to improve the propulsion performance as well as the maneuverability [8,9].

Because of global warming, achieving carbon neutrality requires a global cooperative effort to curb greenhouse gas emissions. As a representative activity, the International Maritime Organization (IMO) has been limiting the operation of ships through the Energy Efficiency Design Index (EEDI) for new ships since 2013 [10–12]. In addition, recently, new indices such as EEXI (Energy Efficiency eXisting ship Index) and CII (Carbon Intensity Indicator) are being introduced for ships currently in operation [13–15]. Accordingly, demand for eco-friendly ships is increasing.

The Asymmetric Pre-Swirl Stator (APSS), one of the most representative Energy Saving Devices (ESDs), is used in various types of ships. The APSS is in front of the propeller and improves efficiency by recovering energy loss caused by the rotation of the propeller. The wake flow on the port and starboard sides varies depending on the direction of rotation of the propeller and the upward flow of the stern. Therefore, an asymmetry in the number of blades (three blades on the port side and one blade on the starboard side) is introduced, which has several advantages [9,16–18]. In this study, the effect of an APSS with rudders on maneuverability was evaluated. In terms of the turning ability, the port turn is different from the starboard turn because the upper part is larger than the lower part from the center point of the propeller. An APSS straightens the flow behind the propeller to the rudder, which makes the turning ability more symmetrical. Jin et al. [18] investigated the flow behind the propeller and found that the tangential flow behind the propeller became straight owing to the optimized APSS. In addition, Su et al. [19] performed a CFD analysis of the PSS and rudder bulb applied to the full scale of a 25 m sized ship, and it was confirmed that the flow field behind the rudder was uniformly improved when an ESD was applied. A study by Koushan, K. et al. [20] compared changes in propeller wake components due to PSS using CFD and experiments where the tangential velocity component greatly increases due to the rotation of the propeller, which has a negative effect on maneuvering performance, and this is partially recovered by PSS.

As previously mentioned, many studies have been conducted on the efficiency improvement of using ESDs; however, only a few studies have examined the maneuverability achieved with such devices. Takekuma et al. [16] studied the maneuverability of a symmetric 6-blade stator and reported that there is no significant difference in maneuverability with and without a stator. Mewis et al. [21] conducted another study on maneuverability using the Mewis duct and observed an improved maneuvering performance when the Mewis duct was used. Kim et al. [22] also examined the maneuverability of an APSS for a tanker and observed an improvement in the maneuverability when as APSS was used, particularly at a more symmetric performance between the port and starboard turns. Kim [23] investigated the course-keeping capability of a tanker with an APSS. The overshooting angle was clearly smaller (improved) with an APSS.

Recently, research on collision avoidance of autonomous ships and improvement of maneuvering performance using special rudders (high-lift rudders) has also been actively conducted. Yim [24] proposed a method to evaluate the effect of turning characteristics on collision avoidance in autonomous ships. Additionally, Kim et al. [25] studied the effect of increased rudder force on the ship's maneuvering performance which was evaluated through numerical simulation. It was confirmed that the increase in rudder lift caused by the high-lift rudder improved the turning circle performance of the ship.

The aim of this study is to determine the effect of ESDs (APSS and bulb) on maneuverability when the newly developed wavy rudder [9] was used. The experiments in this study were conducted in a towing tank at Pusan National University. The rudder forces are the primary focus of the experiments in this study. A more precise prediction of

maneuverability was performed with these data to compare each combination case. In the maneuvering simulation in existing studies, the hull, propeller, and rudder are separated to obtain the hydrodynamic force as an independent part, and statistical coefficients are mainly used for the mutually influencing part. A maneuvering analysis is performed by solving the equations of motion of the rigid body using the hydrodynamic force obtained from this approach. In this study, the mutual interference effects of the hull, ESD, propeller, and rudder were determined via experiments. A maneuvering simulation was conducted by measuring the rudder force according to variations in the rudder angle, considering mutual interference effects. Using this method, a more accurate maneuvering prediction can be obtained by considering the mutual interference effect, which is validated by comparing the results with those obtained from computational fluid dynamics (CFD) analyses. The findings of this study will be useful in the development of more reliable autonomous maritime surface ships.

## 2. Materials and Methods

### 2.1. Principal Dimensions of the Target Ship, Propeller, Stator, and Rudder

The target ship and propeller are the well-known KCS (3600 TEU KRISO Container Ship) and KP505, respectively. The main dimensions of the KCS are shown in Table 1. An APSS was designed for the KCS in a previous study [8]. The principal dimensions and profiles of the proposed stator are presented in Table 2 and Figure 1, respectively. Two rudders (a full-spade rudder (FSR) and a bulb-type wavy twisted rudder (WTR_Bulb) [8]) were used in this study.

**Table 1.** Principal dimensions of the target ship (KCS) and its propeller.

|  | **Full Scale** | **Model Scale** |
|---|---|---|
| Scale Ratio | 39.5 | |
| Length PP [m] | 230.00 | 5.82 |
| Length WL [m] | 232.5 | 5.88 |
| Breadth [m] | 32.26 | 0.81 |
| Depth [m] | 19.00 | 0.48 |
| Design Draught [m] | 10.80 | 0.27 |
| Design Speed | 19.0 knots | 1.56 m/s |
| Propeller Diameter [m] | 7.90 | 0.22 |
| Number of Blades | 5 | |
| Pitch Ratio (0.7R) | 0.997 | |
| Hub Ratio | 0.18 | |

**Table 2.** Principal dimensions of the APSS.

| **Blade No.** | **Position (deg)** | **Pitch Angle (deg)** |
|---|---|---|
| 1st | 45 | 5 |
| 2nd | 90 | 10 |
| 3rd | 135 | 2 |
| 4th | 270 | 1.5 |

The FSR and WTR_Bulb profiles and principal dimensions are presented in Figure 2 and Table 3, respectively. The FSR, which is the reference for comparing the research results, has the same specifications as the semi-spade rudder provided by KRISO, and the skeg part attached to the hull is filled.

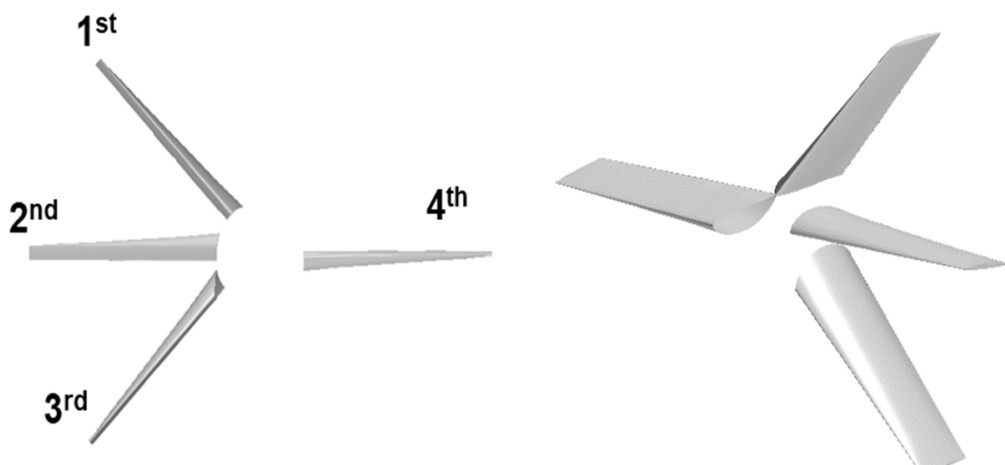

**Figure 1.** Profile of the designed APSS.

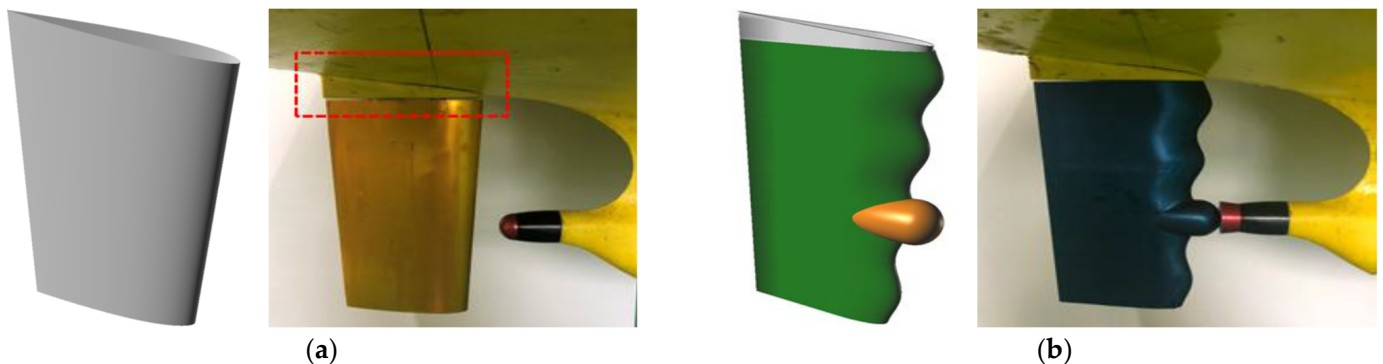

(**a**)　　　　　　　　　　　　　　　　　　　　　　(**b**)

**Figure 2.** Profiles of rudders [8]: (**a**) FSR (Red dotted box: Skeg changed from the official semi-spade rudder); (**b**) WTR_Bulb.

**Table 3.** Principal dimensions of the rudders.

|  | Full-Spade Rudder | Bulb-Type Wavy Twisted Rudder |
|---|---|---|
| Top Chord (m) | 151.90 | 154.59 |
| Bottom Chord (m) | 126.58 | 122.43 |
| Span (m) | 250.63 | 251.71 |

The propulsion efficiency of a combination of ESDs was evaluated in a previous study [9], in which APSS+FSR and APSS+WTR_Bulb improved the propulsion efficiency by 4% and 5.4%, respectively. Figure 3 shows the flow field analyzed using CFD at the 1.017 LPP (Length between Perpendiculars) position of the rudder (just behind the rudder) for each case [9]. The port-to-starboard axial velocity balance improved with the APSS and a higher tangential velocity was recovered compared with the case without the APSS. The axial flow retardation and nonuniformity in the hub area also improved with the WTR_Bulb. These flow phenomena provide a reasonable understanding of the efficiency improvement with the APSS and WTR_Bulb. Maneuverability may also correlate closely with these phenomena, which is the motivation for this study.

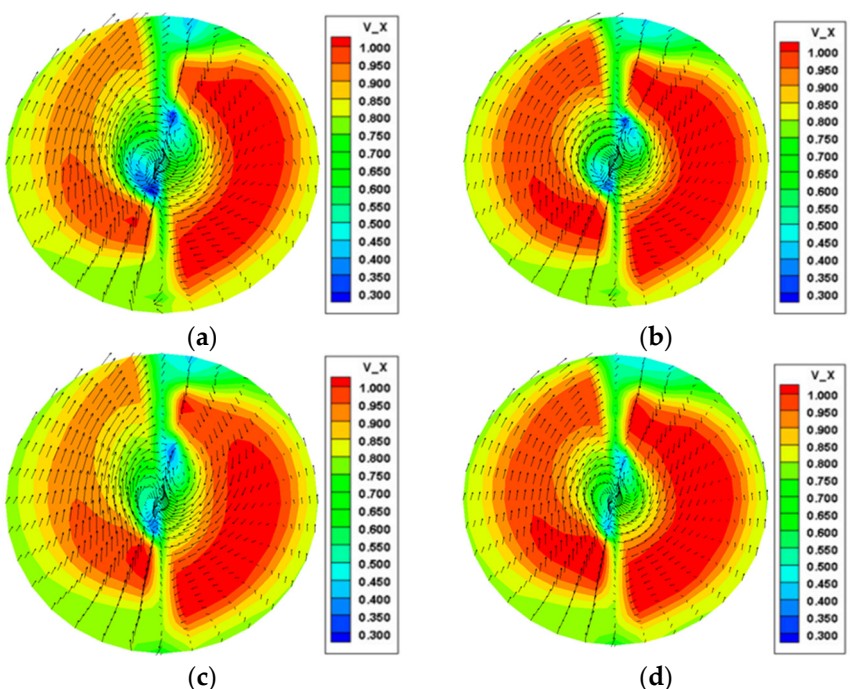

**Figure 3.** Wake distribution [9]: (**a**) FSR; (**b**) FSR+APSS; (**c**) WTR Bulb; and (**d**) WTR Bulb+APSS.

### 2.2. Model Test

The model test was conducted in a towing tank at PNU (L × B × D; 100 m × 8 m × 3.5 m) based on Froude's law of similarity. The hull, propeller, and ESD used in the experiment are shown in Figure 4. The rudder force was measured while rotating the rudder angle in the port and starboard directions within a range of −40–+40° at every increment of 10°, including ± 5° and 35°, at the design speed. The self-propulsion point for each rudder case and the experimental conditions for each rudder case are shown in Table 4. The angle was rotated precisely using an angle controller, and the lift, drag, and torque of the rudders were measured using a rudder force measurement instrument [8]. Figure 5 shows the force and coordinate system acting on the rudder.

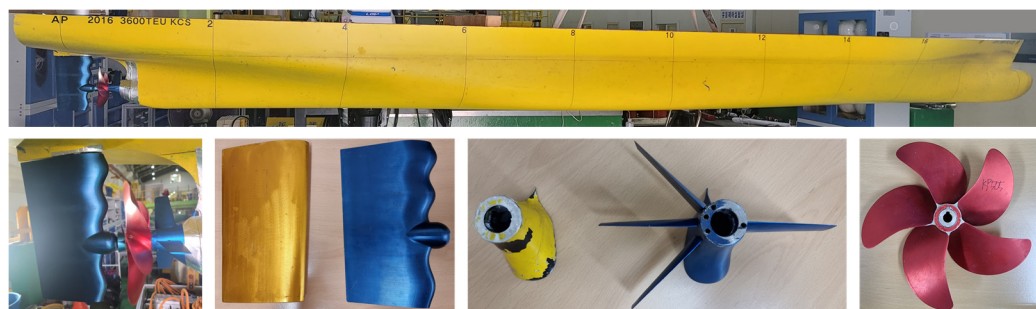

**Figure 4.** Experiment setup with models of the hull, rudder, stator, and propeller.

**Table 4.** Test cases and conditions for resistance, self-propulsion, and rudder force tests.

| Rudder Type | APSS (Attached or Not) | $V_S$ (Knots) | $V_M$ (m/s) | Self-Propulsion Point (RPM) | Rudder Angle (Deg) |
|---|---|---|---|---|---|
| FSR | w/o APSS | | | 659 | −40, −35, −30, −20, −10, −5, 0, 5, 10, 20, 30, 35, 40 |
| | w/ APSS | 24.0 | 1.964 | 637 | |
| WTR_Bulb | w/o APSS | | | 656 | |
| | w/ APSS | | | 635 | |

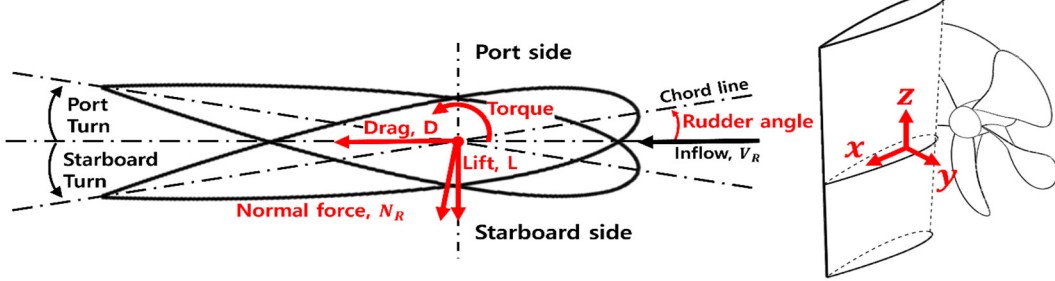

**Figure 5.** Diagram of rudder-induced forces and definition of rudder direction.

*2.3. Model Test Result*

The lift force of the rudder is very important for maneuverability and is normally linearly proportional to the turning force. Figures 6 and 7 show the lift coefficient and drag coefficient according to the angle variation of each case. Each coefficient was nondimensionalized using the formula below.

$$C_L = \frac{L}{\frac{1}{2}\rho V_{0.7r}^2 S} \tag{1}$$

$$C_D = \frac{D}{\frac{1}{2}\rho V_{0.7r}^2 S} \tag{2}$$

In Equations (1) and (2), $V_{0.7r}^2$ is calculated as $\sqrt{V_m^2 + (0.7\pi nD)^2}$ as the speed at the propeller 0.7r. $V_m$ is the velocity of the model ship, $n$ is the RPM of the self-propulsion point, and $S$ is the wetted surface of the rudder.

Overall, the WTR_Bulb is superior to the FSR, as shown in Figure 6. In a conventional FSR, the lift at +turn (port turn) is typically higher than that at −turn (starboard turn). Because the upper part of the rudder from the propeller center is larger than the lower part, the rudder force of the upper part is larger than that of the lower part owing to the propeller's one-directional rotation (top position: propeller-induced tangential vel. From port to starboard; bottom: reverse). The WTR_Bulb used in this study makes this non-symmetric lift force between the port and the starboard almost symmetric by twisting the leading edge of the upper part to the starboard side of 5°, which accordingly makes the angle of attack smaller in the starboard turn case, as shown in Figure 6 where the unbalanced lift at zero rudder angle is shifted to almost zero lift with the proposed devices. The stator also improves the lift and non-symmetric lift; however, the effect is not significant compared with that of WTR_Bulb.

Furthermore, the drag is slightly larger for the WTR_Bulb and APSS cases than for the FSR case. In the WTR_bulb case, the drag around the leading edge is larger than that in the non-twisted case because of the larger angle of attack of the upper part in the starboard turn and, conversely, the larger angle of attack of the lower part in the port turn. Another factor responsible for the increase is the bulb, which leads to added drag. Additionally, as the stator and rudder straighten the rotational flow, the actual flow becomes slightly faster; however, the drag coefficient is nondimensionalized by the same ship speed.

The torques of the various rudder cases were also compared, as shown in Figure 8. The torque of WTR_Bulb+APSS is the lowest, whereas that of FSR is the highest. Although the lift and drag are larger in the WTR_Bulb case, the bulb separation at the leading edge balances the torque increase caused by the trialing edge separation, which is also minimized by the wavy configuration, as noted by Shin et al. [9]. The torque of FSR at zero rudder angle also shifted to zero torque with WTR_Bulb+APSS, similar to that in the lift case. The WTR_Bulb with APSS has some advantages in terms of torque, which leads to a small steering gear capacity and less structural burden.

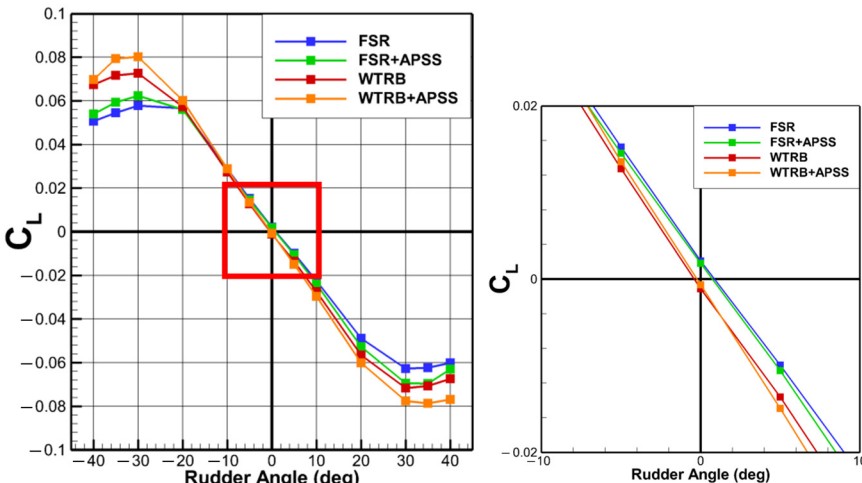

**Figure 6.** Comparison of the rudder lift coefficients for various cases according to variations in the rudder angle. (Right: Enlarged red box).

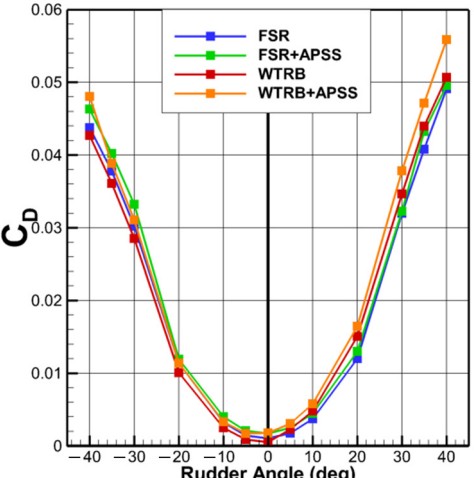

**Figure 7.** Comparison of the rudder drag coefficients for various cases according to variations in the rudder angle.

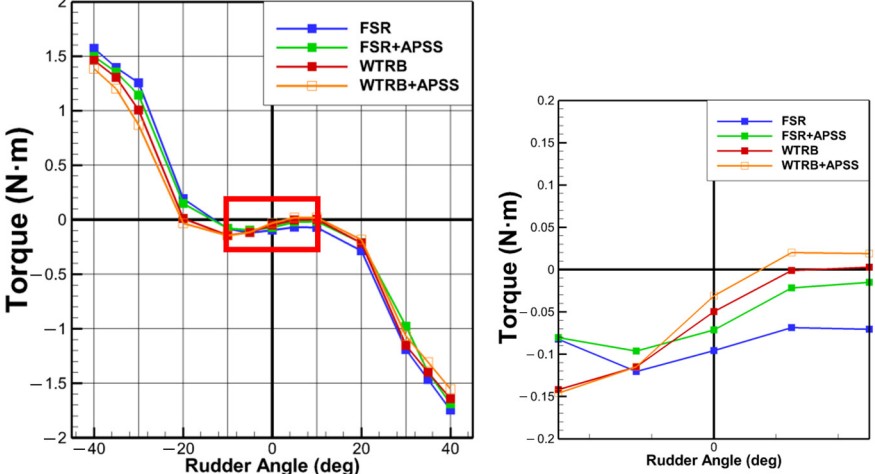

**Figure 8.** Comparison of the rudder torques of various cases according to variations in the rudder angle. (Right: Enlarged red box).

### 2.4. Maneuvering Simulation

#### 2.4.1. Coordinate System

Figure 9 shows a 3-DOF (3 degrees of freedom) ship steering motion (surge, roll, and yaw) in the horizontal plane, where two right-hand coordinate systems are defined, one global and one local.

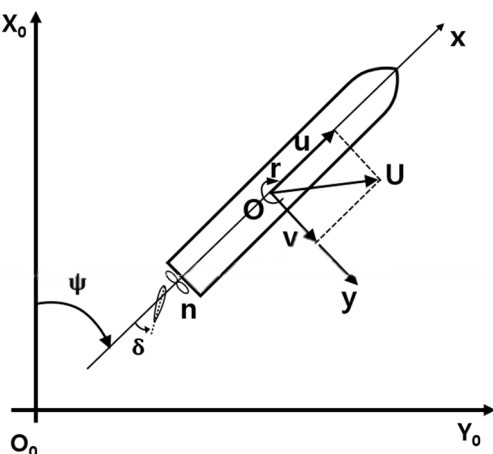

**Figure 9.** Maneuvering coordinate systems.

#### 2.4.2. Motion Equations

The maneuvering modeling group (MMG) standard model proposed by Yasukawa and Yoshimura [26] is given as

$$X = (m + m_x)\dot{u} - (m + m_y)v_m r - x_G m r^2 \tag{3}$$

$$Y = (m + m_y)\dot{v}_m + (m + m_x)ur + x_G m\dot{r} \tag{4}$$

$$N_m = \left(I_{zG} + x_G{}^2 m + J_z\right)\dot{r} + x_G m\left(\dot{v}_m + ur\right) \tag{5}$$

#### 2.4.3. Rudder Forces on Rotating Propeller

The hydrodynamic forces acting on the rudder can be expressed as [26]

$$X_R = -(1 - t_R)F_N \sin\delta \tag{6}$$

$$Y_R = -(1 + a_H)F_N \cos\delta \tag{7}$$

$$N_R = -(X_R + A_H x_H)F_N \cos\delta \tag{8}$$

Here, $t_R$ is the steering resistance reduction factor, $a_H$ is the rudder force increase factor, $X_R$ is the longitudinal coordinate of the rudder position, and $x_H$ is the acting point of the additional lateral force due to the rudder deflection.

The rudder normal force is expressed as

$$F_N = (1/2)\rho A_R U_R{}^2 f_\alpha \sin\alpha_R \tag{9}$$

In this equation, $A_R$ is the profile area of the movable part of the rudder, $U_R$ is the resultant inflow velocity to the rudder, and $f_\alpha$ is the rudder lift gradient coefficient.

$$U_R = \sqrt{u_R{}^2 + v_R{}^2} \tag{10}$$

$$f_\alpha = \frac{6.13\Lambda}{\Lambda + 2.25} \tag{11}$$

In Equation (11), $\Lambda$ is the rudder aspect ratio and $\alpha_R$ is the effective inflow angle.

$$\alpha_R = \delta - tan^{-1}\left(\frac{v_R}{u_R}\right) \tag{12}$$

Here, $u_R$ represents the longitudinal and lateral velocities and $v_R$ is the lateral velocity.

$$u_R = \varepsilon u(1 - w_P)\sqrt{\eta\left\{1 + \kappa\left(\sqrt{1 + \frac{8K_T}{\pi J_P^2}} - 1\right)\right\}^2 + (1 - \eta)} \tag{13}$$

$$v_R = U\gamma_R\beta_R \tag{14}$$

where $\varepsilon$ is the ratio of the wake fractions at the propeller and rudder positions, $\eta$ is the ratio of the propeller diameter to the rudder height, $\kappa$ is a constant, $\gamma_R$ is the flow straightening coefficient, and $\beta_R$ is the effective inflow angle to the rudder in maneuvering motion.

Because of the asymmetrical flows on the port and starboard sides, $\gamma_R$ takes different magnitudes for positive and negative $\beta_R$. $\beta_R$ is expressed as

$$\beta_R = \beta - l'_R r' \tag{15}$$

In this equation, $l'_R$ is the non-dimensional effective longitudinal coordinate of the rudder position.

In a previous study, the above equations were used for the rudder forces to obtain the three-dimensional force (in the x, y, z, directions) [27]. Additionally, the nonsymmetric force of the rudder in the port and starboard directions produced by the propeller can be determined using these equations. In this study, the rudder forces were directly measured under self-propulsion conditions according to the rudder angle variations instead of using statistical coefficients. Simulation of the maneuvering performance was conducted using these measured rudder forces. The hydrodynamic force produced by the KCS hull and propeller acting on the maneuverability was used with existing well-known experimental values which are shown in Tables 5 and 6 [27].

**Table 5.** Maneuvering coefficients for KCS.

| Coefficient | Value | Coefficient | Value | Coefficient | Value |
|---|---|---|---|---|---|
| $Y_v$ | $-0.2591$ | $Y_r - (m + m_x)$ | $-0.1753$ | $Y_{vvr}$ | $-0.4444$ |
| $Y_{vvv}$ | $-1.7212$ | $Y_{rrr}$ | $-0.0228$ | $Y_{vrr}$ | $-0.5461$ |
| $N_v$ | $-0.1421$ | $N_r$ | $-0.0462$ | $N_{vvr}$ | $-0.7339$ |
| $N_{vvv}$ | $-0.2666$ | $N_{rrr}$ | $-0.0313$ | $N_{vrr}$ | $-0.0570$ |
| $(m + m_y)$ | $0.3702$ | $X_{vr} + (m + m_y)$ | $0.3099$ | | |

**Table 6.** Maneuvering coefficients for KCS rudder.

| Coefficient | Value | Coefficient | Value |
|---|---|---|---|
| $(1 - t_R)$ | $0.6715$ | $\kappa$ | $0.3409$ |
| $(1 + a_H)$ | $1.2299$ | $\gamma(\beta_R \geq 0)$ | $0.5218$ |
| $(x_R + a_H x_H)$ | $-0.6168$ | $\gamma(\beta_R < 0)$ | $0.3151$ |
| $\epsilon$ | $1.4391$ | $f_a$ | $2.7244$ |

### 2.4.4. Validation of the Maneuvering Simulation

The validation was performed by comparing our maneuvering simulation results with those of the hydrodynamic coefficients reported in previous studies. This study used the MMG model, and the validation was performed through comparison of the turning circle and zigzag tests.

A comparison of the turning circle test result shows that the turning radius used in the simulation tends to be slightly larger than that of the Hyundai Maritime Research Institute

(HMRI) experiment and almost the same as that of the HMRI CFD result, as shown in Figure 10. The simulation results using the empirical formula are significantly different from those of others [27].

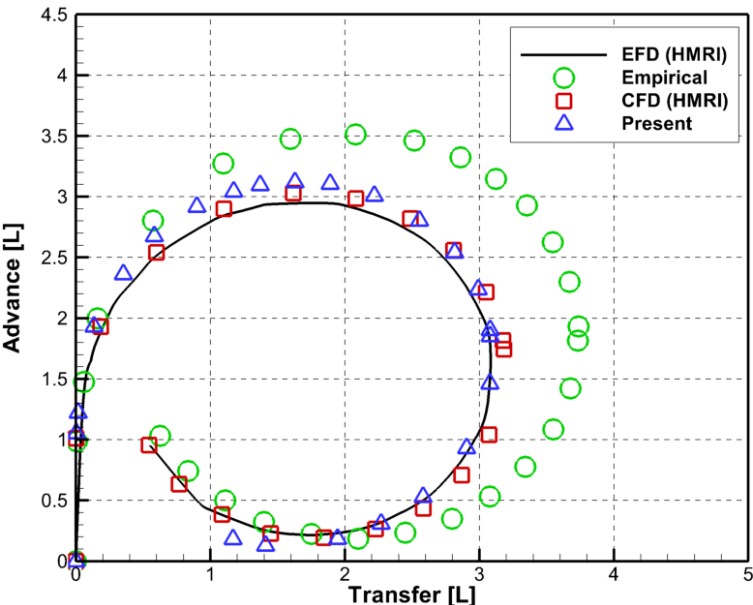

**Figure 10.** Comparison of the turning circles for various cases.

In the 10/10 zigzag test results, there is good agreement between the simulation result of the present study and that of HMRI CFD, whereas the HMRI EFD result is different from those of other methods after the 1st overshoot angle, as shown in Figure 11. The empirical results are significantly different from those of other methods, similar to the turning-circle case. In the 20/20 zigzag test, there is good agreement among all the results, even for the empirical case, as shown in Figure 12.

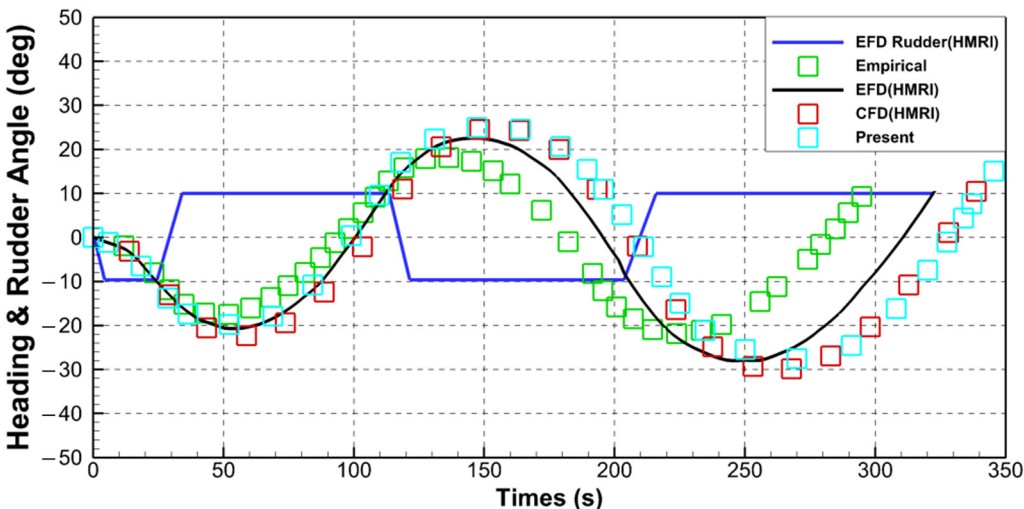

**Figure 11.** Comparison of 10/10 zigzag test overshoot angles for verification of MMG model.

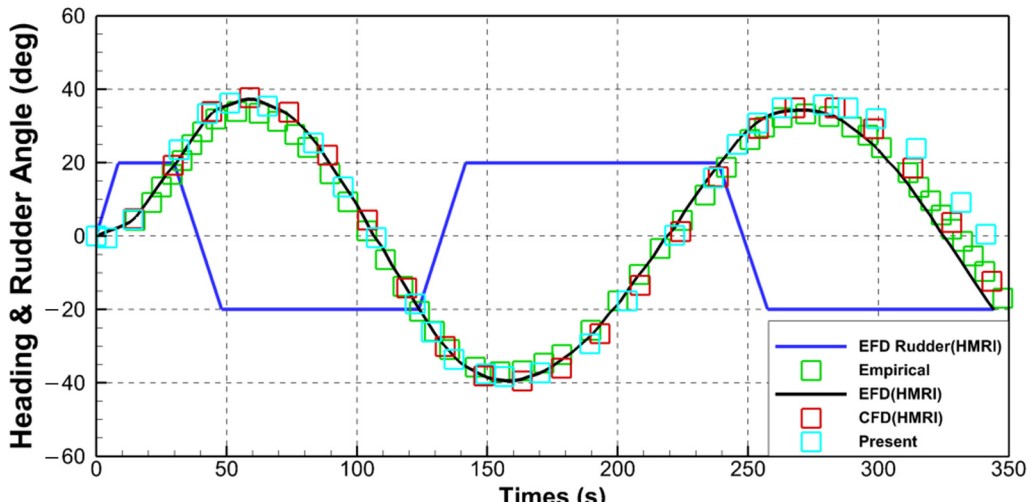

**Figure 12.** Comparison of 20/20 zigzag test overshoot angles for verification of MMG model.

The above comparisons demonstrate the reliability of the simulation in this study.

## 3. Maneuvering Simulation Results

### 3.1. Turning Circle Test

As previously mentioned, the lift force of the rudder on the port turn is larger than that on the starboard turn owing to the propeller's one-directional rotation; accordingly, the turning circle on the port turn is smaller than that on the starboard turn, as shown in Figure 13. Because of the larger rudder force obtained with WTR_Bulb, the turning circle becomes smaller; furthermore, the stator also makes the turning circle smaller, although its contribution is not very large. In addition, the turning circle on the starboard side is much smaller than that on the port side because the twisted upper part from the propeller center is larger than that of the lower part. As a result, the circle of the starboard and port sides becomes almost the same (difference of less than 1.5%) as that obtained with WTR_Bulb. For a more quantitative comparison of turning circles, Table 7 shows the advance length and tactical length along with the difference. And the average of four values {advance(port) + advance(starboard) + tactical(port) + tactical(starboard)}/4 was determined, as presented in Table 8. WTR_Bulb clearly makes the turning circle much smaller than APSS, although the stator also contributes positively.

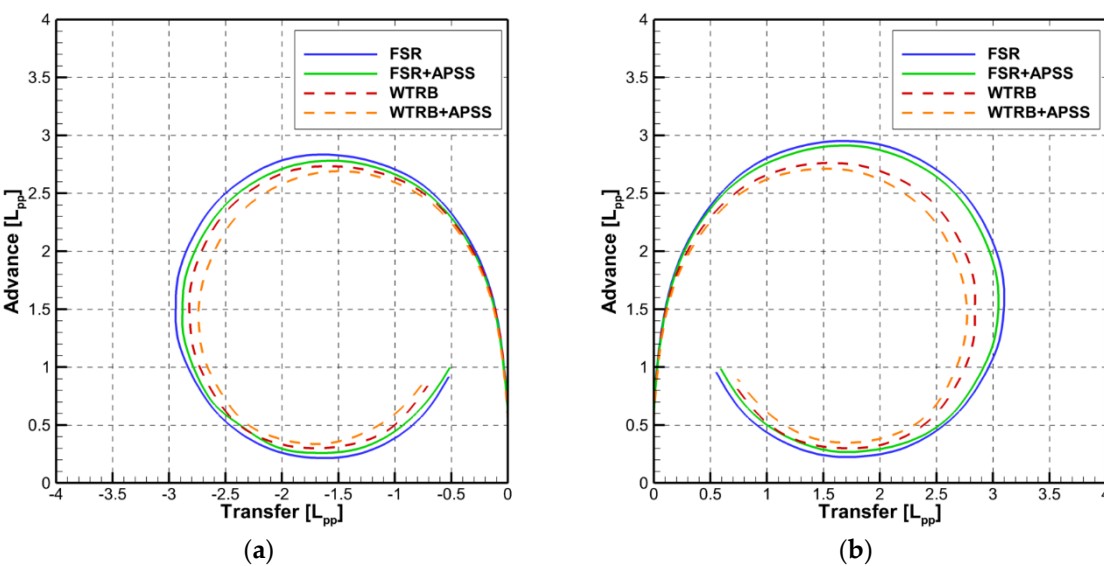

**Figure 13.** Comparison of the turning circles for various cases: (**a**) port turn; (**b**) starboard turn.

**Table 7.** Comparison of advance and tactical diameters for various cases.

| Case | Type | Advance (L) | | | Tactical (L) | | |
|---|---|---|---|---|---|---|---|
| | | Port Turn | Starboard Turn | Diff. (%) | Port Turn | Starboard Turn | Diff. (%) |
| 1 | FSR | 2.83 | 2.95 | 4.24 | 2.94 | 3.10 | 5.44 |
| 2 | FSR+APSS | 2.78 | 2.91 | 4.68 | 2.88 | 3.05 | 5.90 |
| 3 | WTR_Bulb | 2.73 | 2.76 | 1.10 | 2.82 | 2.84 | 1.43 |
| 4 | WTR_Bulb+APSS | 2.69 | 2.71 | 0.74 | 2.74 | 2.77 | 1.09 |

**Table 8.** Comparison of the average turning circles for various cases.

| Case | Type | Average Turning Circle * | Improvement (%) |
|---|---|---|---|
| 1 | FSR | 2.96 | reference |
| 2 | FSR+APSS | 2.91 | 1.7 |
| 3 | WTR_Bulb | 2.79 | 5.7 |
| 4 | WTR_Bulb+APSS | 2.73 | 7.7 |

* Average turning circle: {advance (port + starboard) + tactical (port + starboard)}/4.

### 3.2. Zigzag Test

The trend of maneuverability in the zigzag test is similar to that in the turning circle case. As shown in Figures 14 and 15 and Table 9, the overshoot angles in the 10/10 and 20/20 zigzag tests with WTR_Bulb are smaller, and the angle is much smaller on the starboard turn than on the port turn, which is almost the same as in the turning circle case. From a quantitative point of view, the improvement with WTR_Bulb is significantly large, as presented in Tables 10 and 11. Although a slight decrease is observed in the overshoot angle of the 10/10 zigzag test with APSS, a significant synergistic effect is observed with the combination of WTR_Bulb and APSS, as presented in Table 9. In the 10/10 zigzag test, the increase in the 2nd overshoot angle is larger than that in the 20/20 zigzag test, as presented in Table 10. The devices used in this study (FSR, WTR_Bulb) particularly decreased the 2nd overshoot angle in the 10/10 zigzag test, which is significantly decreased by the combination of WTR_Bulb and APSS, as presented in Table 10. The 2nd overshoot is normally difficult to control because the momentum of the rotation increases after the 1st overshoot angle. The devices used in this study are significantly more effective for this maneuverability because the rudder performance, such as lift and balance, is significantly better than that of conventional rudders. The best case is always that of WTR_Bulb combined with APSS, as presented in Tables 9–11.

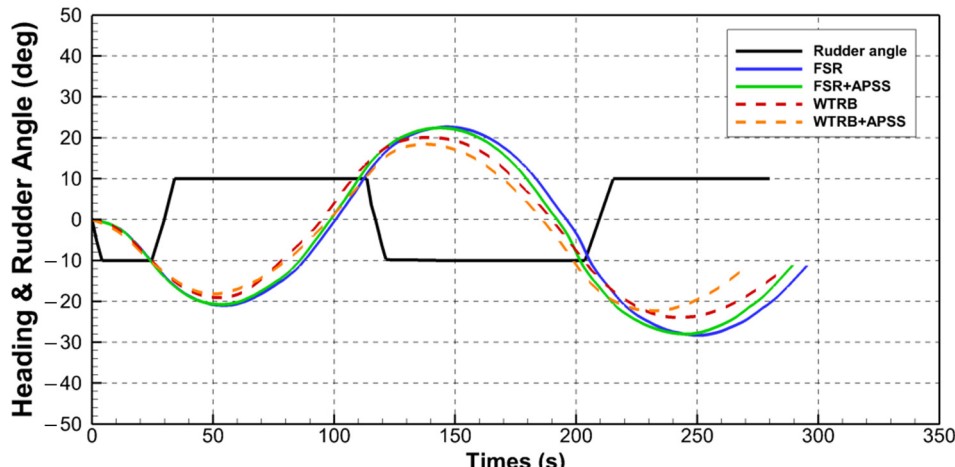

**Figure 14.** Comparison of the overshoot angles for various cases in the 10/10 zigzag test.

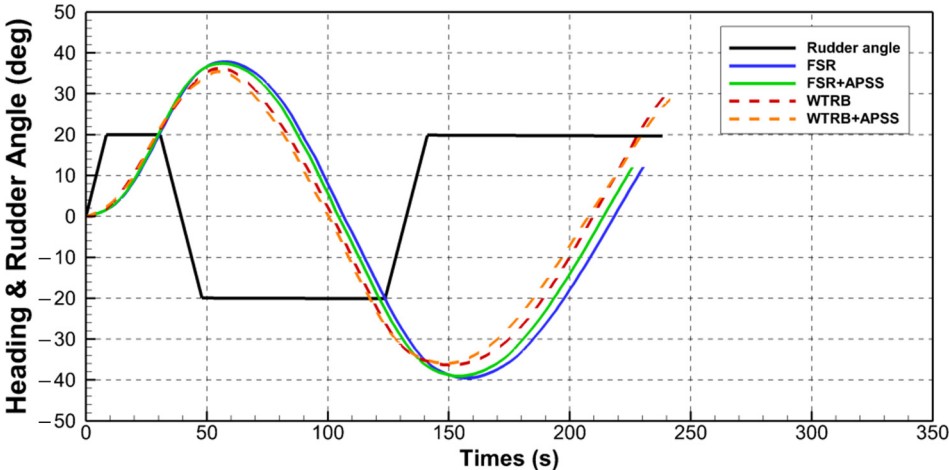

**Figure 15.** Comparison of the overshoot angles for various cases in the 20/20 zigzag test.

**Table 9.** Comparison of the overshoot angles for various cases in the 10/10 and 20/20 zigzag tests.

| Case | Type | 10/10 Zig-Zag | | | 20/20 Zig-Zag | | |
|---|---|---|---|---|---|---|---|
| | | 1st O.S.A * | 2nd O.S.A | Diff. (%) | 1st O.S.A | 2nd O.S.A | Diff. (%) |
| 1 | FSR | 10.84 | 12.58 | 16.05 | 17.78 | 19.47 | 9.51 |
| 2 | FSR+APSS | 10.64 | 12.34 | 15.98 | 17.40 | 18.98 | 9.08 |
| 3 | WTR_Bulb | 8.87 | 10.05 | 13.30 | 15.51 | 16.27 | 4.90 |
| 4 | WTR_Bulb+APSS | 8.08 | 8.39 | 4.61 | 15.26 | 15.76 | 3.28 |

* O.S.A: Overshoot Angle.

**Table 10.** Comparison of the average overshoot angles for various cases in the 10/10 zigzag test.

| Case | Type | Average Overshoot Angle * | Improvement (%) |
|---|---|---|---|
| 1 | FSR | 11.71 | reference |
| 2 | FSR+APSS | 11.49 | 1.9 |
| 3 | WTR_Bulb | 9.46 | 19.2 |
| 4 | WTR_Bulb+APSS | 8.24 | 29.7 |

* Average overshoot angle: (1st + 2nd)/2.

**Table 11.** Comparison of the average overshoot angles for various cases in the 20/20 zigzag test.

| Case | Type | Average Overshoot Angle * | Improvement (%) |
|---|---|---|---|
| 1 | FSR | 18.63 | reference |
| 2 | FSR+APSS | 18.19 | 2.3 |
| 3 | WTR_Bulb | 15.89 | 14.7 |
| 4 | WTR_Bulb+APSS | 15.51 | 16.7 |

* Average overshoot angle: (1st + 2nd)/2.

## 4. Conclusions

In this study, ESDs (APSS and Bulb) were applied to a developed wavy twisted rudder. The maneuverability of these devices was investigated. Experimental studies were conducted to compare the maneuverability with that of an FSR. In previous studies [9], the developed wavy twisted rudder was compared with a conventional twisted rudder, and it was reported that the rudder force is larger, whereas the propulsion performance, that is the efficiency, is almost the same but slightly worse. To increase the efficiency and maneuverability, the ESDs of the asymmetric pre-swirl stator and bulb were applied to the developed wavy twisted rudder.

The performance of the rudder with these ESDs, such as lift, drag, and torque, was experimentally investigated based on the efficiency increase obtained with WTR_Bulb and

APSS. Stall delay and starboard–port balancing effects are clearly observed with WTR_Bulb, and APSS also contributes to these functionalities, although its contribution is not large compared with that of WTR_Bulb.

A maneuvering simulation, including the mutual interference effects of the hull, ESD, propeller, and rudder, was conducted by measuring the rudder force. The method used in this study is based on the MMG model, and it was validated by comparing the result with those obtained using the EFD and CFD of the HMRI. The simulation results of the turning circle and zigzag tests obtained using the proposed method agree fairly well with those of the HMRI. Four cases (a combination with and without APSS and WTR_Bulb) were adopted to compare the maneuverability performance. For the turning circle, 10/10 zigzag, and 20/20 zigzag, significant improvements were observed with WTR_Bulb and APSS, although the effect is not large compared with that with WTR_Bulb. These devices, particularly the WTR_Bulb, play a significant role in decreasing the turning circle and overshoot angle, as well as producing a symmetric behavior of the port and starboard turn. Although APSS works quietly compared with WTR_Bulb in maneuverability, combining APSS and WTR_Bulb significantly improved the overshoot angle in the 10/10 zigzag test.

In the future, we will use CFD analyses to validate the performance obtained in this study in terms of maneuverability and propulsion. The developed WTR_Bulb with APSS could be useful in the development of a more reliable maritime autonomous surface ship.

**Author Contributions:** Conceptualization, M.C.K.; validation, Y.-J.S. and J.W.K.; investigation, Y.-J.S. and K.-W.L.; resources, M.C.K.; writing—original draft preparation, Y.-J.S.; writing—review and editing, J.W.K. and M.C.K.; visualization, W.S.J.; supervision, M.C.K.; project administration, Y.-J.S.; funding acquisition, M.C.K. All authors have read and agreed to the published version of the manuscript.

**Funding:** This research was funded by the Korea Institute of Energy Technology Evaluation and Planning (KETEP) and the Ministry of Trade, Industry and Energy (MOTIE) of the Republic of Korea (20224000000090), and was also supported by the Basic Science Research Program through the National Research Foundation of Korea (NRF) funded by the Ministry of Science, ICT and Future Planning (NRF-2022R1F1A1070993).

**Institutional Review Board Statement:** Not applicable.

**Informed Consent Statement:** Not applicable.

**Data Availability Statement:** Not applicable.

**Conflicts of Interest:** The authors declare no conflict of interest.

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
