# Peer review of "Maneuverability Performance of a KRISO Container Ship (KCS) with a Bulb-Type Wavy Twisted Rudder and Asymmetric Pre-Swirl Stator"

_jmse, doi:10.3390/jmse11102011_

Round 1

Reviewer 1 Report

Taking KCS as an example, the authors experimentally investigated the rudder performances of an energy-saving device combination, a bulb-type wavy twisted rudder and asymmetric pre-swirl stator and demonstrates the effects of such energy-saving devices on ship maneuverability. Conclusion indicates that the adoption of the bulb-type wavy twisted rudder and asymmetric pre-swirl stator could improve the maneuvering performances. While the paper has merits, improvements are needed for better reading experience:

1. In the first section, the introduction of background, '', seems to have little correlation with the subject. The authors did not indicate the relationship between the ship maneuverability, ship collision avoidance, and autonomous ship. It is suggested to make some supplement, or cut to the chase, indicating that the special rudders are developed to improve the ship maneuverability and efficiency.

2. In figure 4, a subfigure illustrating the rudder, stator, and propeller of the stern section is needed.

3. In figure 6, the measured data and tendency line is covered by the legend.

4. In table 5, maneuvering coefficients related to the longitudinal forces are missed.

5. In table 8, the right bracket of the definition of the average turning circle needs to be adjusted.

6. As the asymmetric pre-swirl stator and the bulb-type wavy twisted rudder are adopted, the thrust deduction factor, wake coefficients at the propeller and rudder positions have slight differences. I believe assumption that these factors are the same is made in the present paper, the authors should make an explanation.

 Minor editing of English language required.

Reviewer 2 Report

The paper carried out the relevant research by means of CFD and model experiment, and obtained the expected results, which is worthy of affirmation for the whole work.

From the whole paper, CFD simulation is a very important part. Usually, the maneuverability verification of CFD of the whole ship is complicated and the workload is huge. The curve of the paper shows the comparison curve of CFD. It is suggested to supplement the modeling part of CFD.

Judging from the content of the paper, the English writing is smooth and easy to understand. For me, English is not the first language, it is recommended to find more professional English personnel to optimize.

Author Response

Thank you for your comment. 

In the present study, The measured hydrodynamic forces such as lift, drag and moment were used for the maneuvering simulation (MMG model).

The CFD work for the rudder forces is expected to be conducted in the near future. The CFD mentioned in the present paper is shown in the validation of the maneuverability performance in comparison with Hyundai Marine Research Institute (HMRI)'s CFD and EFD.

Reviewer 3 Report

None

Author Response

Thank you for your review.

Reviewer 4 Report

General

The authors develop and test maneuverability characteristics of a KCS with energy-saving novel bulb-type wavy rudder and an asymmetric pre-swirl stator. For various cases turning circle and zigzag tests were simulated to reveal the effects of stator and bulb-type wavy rudder.

Overall, the work is scientifically sound, well-presented, and the language is good. It should be acceptable after a final check for language and typos. This reviewer recommends only to make a small change in the title by writing KRISO Container Ship in place of KCS. This is a well-known ship in the field of naval architecture but it would be better to make it clear for readers from other areas who may not be familiar with KCS.

Conclusion

According to this reviewer, the manuscript is scientifically sound, well-presented and its language is quite good. The manuscript is acceptable after a final check for overlooked points and typographical errors.

Author Response

Thank you for your comment.

I changed the title based on your comment.

Round 2

Reviewer 1 Report

The explanations of Tables 4,5,6,7 are missing, they are not mentioned in text. Please include them in the final submission. 

I have no further comment on the quality of English Language. 

Author Response

Thank you for commenting on what I missed.

Descriptions for each table have been added.

Thank you.